# Succinylation Improves the Thermal Stability of Egg White Proteins

**DOI:** 10.3390/molecules24203783

**Published:** 2019-10-21

**Authors:** Dabo He, Ying Lv, Qigen Tong

**Affiliations:** 1College of Food Science and Engineering, Beijing University of Agriculture, 7 Beinong Lu, Changping District, Beijing 102206, China; hedabo@bua.edu.cn (D.H.); lvying@bua.edu.cn (Y.L.); 2Beijing Engineering Research Center of Egg Safety Production and Processing, Beijing 100094, China; 3Beijing Laboratory for Food Quality and Safety, Beijing 100022, China

**Keywords:** egg white protein, succinylation, thermal stability, conformational structure, aggregation

## Abstract

Succinylation can improve the thermal stability of various proteins. In this study, succinylated egg white protein (SEWP) samples with different succinylation degrees were prepared by adding various succinic anhydride additives to egg white protein (EWP). The thermal stability of SEWP and the conformational structure under various succinylation degrees were investigated. With the increase in succinylation degree, the turbidity of heated SEWP solution (90 °C for 30 min) markedly declined. The heated SEWP solution with high succinylation degree (37.63%, 66.57%, and 72.37%) was transparent. Moreover, the result of differential scanning calorimetry confirmed that the thermal stability of succinylated EWP increased. The results of intrinsic fluorescence spectra and Fourier-transform infrared spectroscopy illustrate that succinylation changed the conformational structure of EWP. Succinylation increased the electrostatic repulsion and decreased the surface hydrophobicity, and it changed the aggregation morphology of EWP. Cross-linked spherical aggregates of low succinylation degree transformed to thready aggregates of a high succinylation degree. Thus, succinylation improved the thermal stability of EWP.

## 1. Introduction

In recent decades, milk, whey protein, and soy protein isolates have been preferred ingredients in high-protein beverages due to their high nutritional value and production. Egg white is rich in nutrients that are easy to absorb; it is widely applied to various foods due to its low cost. By using egg white protein (EWP), the nutritional content of protein beverages can be increased and their production cost can be decreased. EWP has the characteristic advantages of good gelation properties [1], but during the processing of beverage, gelation properties induce a negative effect: EWP thermal aggregation can bring about tube blocking, a decrease in product quality during the heat treatment of egg white protein, or gel formation after sterilization as a liquid ingredient. The potential for thermal aggregation limits the application of EWP in protein beverages. In the traditional beverage industry, heat treatment plays a key role in improving microbial safety, protein digestibility, and absorption of nutrients [2,3]. Therefore, satisfying the demands of EWP beverage requires improving the thermal stability of EWP. At present, there are few studies regarding improving the thermal stability of EWP. Hong et al. [4] reported that the addition of arginine at 106 mg/mL can effectively control the formation of insoluble aggregates of EWP solution (1 mg/mL); the amount of arginine used far exceeds the content of EWP, and arginine cannot suppress the EWP solution of high-concentration thermal aggregation. According to previous study [5], high-pressure treatment (550 MPa) for 20 min can inhibit the formation of insoluble aggregates in the EWP solution, while high-pressure treatment can have negative effects on rheological properties and color. As reported by Kulchaiyawat, Wang, and Han [6], succinylation by octenyl succinic anhydride can improve the thermal stability of albumen; however, the reason for the improvement of thermal stability has not been further investigated. The succinylation process the advantages of high efficiency and, with the method of high pressure treatment, succinylation will be more practicable and effective when compared with the addition of arginine.

Succinylation is one of the commonly used chemical modification methods that can improve the solubility, emulsifying, and forming properties of proteins [7,8,9,10]. According to the Food and Drug Administration (FDA), succinic anhydride (21 CFR 175.300) is generally recognized as safe (GRAS) for miscellaneous and general purpose in the United States, and there are no other restrictions other than good manufacturing practices [11]. Amino succinylation has been demonstrated to be the main reaction between succinic anhydride and protein. It is a typical nucleophilic substitution reaction that takes place between the ε-amino group of lysine and succinic anhydride [12]. For several years, succinylated modification has been applied to some proteins, such as soy protein isolate, milk protein concentrate, sodium caseinate, yak casein micelles, and oat protein. Succinylation can improve the solubility of milk protein concentrate and sodium caseinate. The result of differential calorimetry scanning illustrated that the stability of milk protein decreased, while the introduction of the succinic group increased the surface hydrophobicity and changed the surface charge state, which, in turn, suppressed the aggregation of protein [13,14]. Succinylation promoted the dissociation of phosphoprotein of yak casein micelles and changed the secondary structure of protein, thus lowering the surface hydrophobicity and heat stability of yak casein micelles [15,16]. Oat protein and soy protein are similar to other proteins, and succinylation can alter the secondary and tertiary structure of protein, which improves the colloid stability and suppresses the aggregation of protein while decreasing the thermal stability of protein [17,18].

Nevertheless, there are few studies regarding the improvement of thermal stability of EWP. Hence, the purpose of this study was to improve the thermal stability of EWP by succinylation. This study investigated the thermal stability by determining the turbidity and the differential thermal analysis curve of EWP. The changes in the surface hydrophobicity, zeta potential, intrinsic fluorescence spectra, Fourier transform infrared spectra, and aggregation morphology of native and succinylated EWP were determined and used to analyze the reasons for the changes in stability of EWP as compared with the control. We hypothesized that succinylation could increase the thermal stability of EWP.

## 2. Results and Discussion

### 2.1. Determination of Succinylation Degree

Succinylation is a frequently used chemical modification method for protein and it is well known to improve the functionality of proteins. Succinylation involves chemical derivatization of ε-amino group of lysine in protein with succinic anhydride [14]. The succinylation degree of native and succinylated EWP refers to the changes in free amino groups [12,14]. Changes in the EWP succinylation degree relative to the increase in the added amount of succinic anhydride were divided into approximately two stages (Figure 1). When the molar ratio of anhydride: amino groups was raised between 0.0 and 2.0, the succinylation degree increased rapidly from 0.00% to 66.57%; when the molar ratio further increased, the succinylation degree slowly increased. Thus, a molar ratio of 2.0 is the critical point of succinylation. Succinylation proceeds from the surface of proteins to the inside of proteins; a critical succinylation degree indicates that the protein structure dramatically changed [19]. This result indicated that, when the molar ratio raised to 2.0, the succinylation degree increased to 66.57%, which indicates that the content of lysine in the surface of EWP was at approximately 66.57%, and with the further increase of molar ratio, the lysine exposed from the inside of proteins further reacted with succinic anhydride. This is similar to the succinylation degrees for soy protein and yak casein micelles with the same additive of succinic anhydride [16,18]. Nonetheless, the succinylation degrees of sodium caseinate were 64.19% and 96.08% when the succinic anhydride: casein molar ratios were 1 and 5, respectively [14], far higher than the succinylation degree of EWP at the same ratio. We speculate that the difference in the succinylation degree was due to the type of proteins and the amount of lysine exposed to the surface of protein. Although globular proteins is the major component of soy protein, casein, and EWP [20,21,22], original casein micelles in milk were destroyed and separated with a mineral substance in the production of casein, which led to casein molecular unfolding and the exposure of a large amount of lysine to react with succinic anhydride [23,24]. Thus, the various content of lysine that was exposed to the surface of proteins probably led to the difference in the succinylation degree of proteins. Furthermore, we found that the succinylation degree of other proteins with a large amount of added succinic anhydride, including soy proteins and casein, could reach up to 90%. However, it was difficult for succinylated egg white protein (SEWP) to exceed 75% degree of succinylation. This phenomenon probably resulted from changes in the structures of soy proteins and casein due to the exposure of lysine groups buried in the interior regions of the protein upon the induction of succinylated groups with further succinylation [18], but with the increase in the amount of added succinic anhydride, there was almost no further exposure of the lysine groups on the surface of EWP molecules. Hence, less succinic anhydride was able to react with the lysine groups of EWP.

### 2.2. Turbidity

Changes in turbidity can reflect the alteration of the thermal stability, the degree of aggregation, and the amount of suspension particles [25,26]. Figure 2 shows the changes in the turbidity of EWP and SEWP solution at various ionic strengths. When compared with the EWP, the turbidity of SEWP significantly decreased (*p* < 0.05). The solutions were transparent as the succinylation degree reached 37.63%, indicating that succinylation suppressed the aggregation and improved the thermal stability of EWP. It is known that NaCl is a very important food additive in the process of food handling, particularly for improving the flavor and shelf life. The addition of NaCl can decrease the amount of hydrosulfonyl exposed on the surface of protein in the process of heat treatment, increase the surface hydrophobicity, and neutralize the charge on the protein surface. These promote the formation of insoluble aggregates and they have a negative effect on the food quality and food processing [27]. Accordingly, the turbidities of EWP and SEWP solutions with different succinylation degrees at various ionic strengths were determined. The turbidity of EWP solution increased with increasing ionic strength and produced precipitate at a high ionic strength. After mild succinylation (2.70% and 10.83%), the turbidity of SEWP solution with the addition of NaCl decreased, and the layering weakened; beyond this value, there was no layering in the SEWP solution with the addition of NaCl. This result suggests that succinylation can prevent the ion-induced aggregation of EWP. This phenomenon is similar to the result for succinylated albumin that was reported by previous research [6].

### 2.3. Differential Scanning Calorimetry (DSC)

DSC could be used to determine the thermal stability of protein. ΔH represents the energy that leads to the unfolding and destruction of the quaternary structure of protein [28]. Figure 3 shows the thermal stabilities of EWP samples of various succinylation degrees. The unheated EWP sample displayed two typical enthalpy transitions, which correspond to the Td of ovotransferrin (67.7 °C) and ovalbumin (82.3 °C) [29]. With the increase in succinylation degree, the enthalpy of denaturation of the two proteins decreased. Succinylation caused peaks that were smaller than those of the control samples, which indicated that less denaturation occurred in SEWP as compared with EWP in heat treatment. These results are consistent with lysozyme, soy proteins, and yak casein micelles, whereby, with increasing succinylation, the Td value and ΔH of these proteins decreased [16,18,30]. This phenomenon might be related to the similarity of succinylation sites of this protein. Moreover, at 73.8% of succinylation degree, succinylation led to the disappearance of peak 1, and the area of peak 2 remarkably decreased. This indicates that, after succinylation of a high degree, ovotransferrin have totally denatured and ovalbumin have partially denatured beforehand, which was probably related to the changes in conformational structure of EWP after succinylation of high degree. This phenomenon resulted from an increase in the thermal stability of SEWP.

### 2.4. FTIR Spectroscopy

FTIR spectroscopy was used to study the conformational structure of EWP and SEWP with different succinylation degrees [31]. Figure 4 illustrates the FTIR spectroscopy of EWP and SEWP with different succinylation degree. The spectra were similar to that of ovalbumin [32], which is the most abundant protein in egg white [21]. There were five major bands in the FTIR spectra for EWP, including amide I, amide II, amide III, amide A, and amide B. Amide I (1600–1700 cm^−1^) is associated with C=O stretching vibrations [33] and the secondary structure of protein [34,35]. The characteristic absorption band of EWP was observed at 1652 cm^−1^ with the increase in the degree of succinylation. The amide I shifts to the low frequency were noticeable at wavenumbers of 1650, 1648, 1648, 1646, 1654, 1646, and 1648 cm^−1^. This shift illustrates that the ordered structures increased [36] and the α-helix transformed to β-sheet [37]. Amide III represent C–N stretching vibration and N–H bending [38]. When the succinylation degree increased, amide III slightly shifted (1240 cm^−1^ for native EWP) to lower wavenumbers: 1240, 1241, 1241, 1243, 1243, 1243, and 1243 cm^−1^ for SEWP, from 2.7 to 72.36% succinylation degree. The shift of amide III band into a higher-frequency domain indicates that the interaction between the –SH and amide group of peptide chains weakened. Amide A represented the vibration stretching of N–H group with hydrogen bonds. With the increase in the succinylation degree, amide A shifted from 3298 cm^−1^ (EWP) to lower wavenumbers: 3297, 3295, 3295, 3293, 3291, 3291, and 3291 cm^−1^. This phenomenon illustrates that the N–H group of peptide chains combined with more hydrogen bonds. Amide B is associated with vibration stretching of the =C–H group and NH_3_^+^. After succinylation, the band of amide B shifted from 2964 cm^−1^ to lower wavenumbers: 2963, 2963, 2963, 3963, 2962, 2962, and 2962 cm^−1^. This shift illustrates the interaction between NH_3_^+^ of the peptide chain and the asymmetric stretching vibration of the –CH_2_ group, as well as NH_3_^+^ of peptide side chains [39]. The observed shifts indicate that succinylation changed the secondary structure, strengthened the hydrogen bond between peptide chains, and changed the conformational structure of EWP.

### 2.5. Surface Hydrophobicity

Figure 5 shows the changes in the surface hydrophobicity of EWP and SEWP. The surface hydrophobicity of SEWP decreased significantly when compared with the EWP (0.0%) (*p* < 0.05). This might have been due to the changes in the molecular structure of EWP after succinylation: The amount of available binding sites for ANS decreased and less of the ANS probe could access and bind to hydrophobic groups [12,40]. Moreover, the combination between the ANS probe and the hydrophobic site mainly depended on the electrostatic force between them [41] with the introduction of succinylated groups. The increased steric resistance effect and electrostatic repulsion had an effect on the combination of the ANS probe and the hydrophobic site [15,42]. Heat treatment destroyed the protein spatial structure and opened the polypeptide chains, leading to the exposure of hydrophobic groups in the interior regions of the protein, as well as more access for protein molecules to make contact with each other and to form large aggregates. However, succinylation could change the conformational structure and decrease the surface hydrophobicity of protein, thus preventing the thermal aggregation and improving the thermal stability of EWP. This result is similar to those of proteins that were reported by other studies, including milk protein concentration, sodium caseinate, and yak casein micelles [13,14,15].

### 2.6. Intrinsic Fluorescence Spectra

Aromatic amino acids, including tryptophan, phenylalanine, and tyrosine, produce fluorescence at a certain excitation wavelength. Tryptophan is generally used as the probe for intrinsic fluorescence. In this study, the intrinsic fluorescence spectra of EWP and SEWP solution at 290 nm were obtained to determine the changes in the tryptophan residue microenvironment and the conformational structure of EWP and SEWP [43]. Figure 6 illustrates the intrinsic fluorescence spectra of EWP and SEWP with different succinylation degrees. The maximum wavelength (λ_max_) of EWP was 333 nm. λ_max_ of SEWP of a low degree (2.70%–37.63%) was unchanged relative to that of EWP, but the fluorescence intensity increased. This result indicates that succinylation changed the structure of EWP. This phenomenon could interpreted by a high content of tryptophan being exposed in polar aqueous solution, leading to spontaneous fluorescence quenching with the introduction of succinic anhydride groups; the conformational structure of proteins changed and the amount of lysine exposed to solution decreased, which weakened the fluorescence quenching, thus the fluorescence intensity increased [44]. With a further increase of the succinylation degree (66.57%–74.83%), the λ_max_ of SEWP increased from 333 nm to 336 nm, red shifts occurred, and the fluorescence intensity decreased. This phenomenon illustrates that the polarity of the microenvironment of the tryptophan residue increased and succinylation destroyed the surface hydrophobic interaction within the protein molecules [18]. During the preliminary stage of succinylation, amino acids on the surface of EWP reacted with succinic anhydride and the introduction of a small amount of negatively succinic anhydride group, resulting in a slight change in the conformational structure of EWP. In addition, tryptophan residues and surface hydrophobic groups on the protein surface became buried in the interior of the protein, which makes it difficult for the succinylated protein to aggregate. When the succinylation degree continued to increase, the introduction of a large amount of succinic anhydride group destroyed the interior structure of EWP and the protein structure dramatically changed. In the DSC thermograms (Figure 3), the enthalpy of SEWP with a low succinylation degree (2.7%–37.63%) decreased slightly when compared with EWP; the enthalpy of SEWP with a higher succinylation degree (66.57%–74.83%) significantly decreased, or disappeared. This further illustrates that succinylation of a low degree changed the structure of EWP slightly and that succinylation of a high degree drastically destroyed the structure of EWP.

### 2.7. Zeta Potential

The zeta potential refers to the potential of the shear layer on the surface of zone particles, which are commonly used to describe the electrostatic force between colloidal particles and the stability of colloids. A zeta potential of 10 mA is the critical value for particle aggregation; a 30-mA zeta potential represents a solution system process of a higher stability [19]. A higher zeta potential represents a stronger repulsive force and lower possibility for the aggregation of particles. There was a strong repulsive force between colloidal particles with large homogeneous charge, which made them resistant to aggregation and maintained the stability of the colloid. Figure 7 shows the changes in zeta potential of EWP and SEWP. Relative to that of EWP, the zeta potential of SEWP generally increased. With the increase in the succinylation degree, the zeta potential increased first and then there was then a slight downward trend. The induction of a succinic anhydride group with negative charge increased the negative charge between protein particles, which produced electrostatic repulsion among EWP particles and prevented the aggregation of EWP. Hence, the results suggest that the inhibition of EWP aggregation is not only related to the decrease in surface hydrophobicity, but also to the increase in electrostatic force. With the increase in the succinylation degree, the turbidity of SEWP continued to decrease (Figure 2), which means that the thermal aggregation of EWP was further suppressed and the thermal stability increased. However, the zeta potential of the SEWP solution with a high succinylation degree (66.57%–73.84%) was reduced. This result suggests that the further increase in the thermal stability of SEWP (66.57%–73.84%) was independent of the electrostatic force, but it was related to the decrease in the surface hydrophobic force (Figure 5). The red shift only occurred in the intrinsic fluorescence spectra (Figure 6) of SEWP with a high succinylation degree (66.57%–73.84%), which indicated that the structure of SEWP with a high degree drastically changed as compared with EWP and that the surface hydrophobic interaction among the SEWP molecules weakened. Succinylation with a high degree induced a large amount of succinic anhydride groups with a negative charge into EWP, and the conformational structure of EWP dramatically changed, which results in a decrease in the electrostatic force on the surface of EWP.

### 2.8. Transmission Electron Microscope (TEM)

In this study, TEM was conducted to observe the aggregation morphology of EWP and SEWP in the protein solution that was heated at 90 °C for 30 min (Figure 8). The EWP formed large, dense, and highly cross-linked aggregates. SEWP of a low succinylation degree (2.70% and 5.83%) formed highly disperse but large aggregates; as the succinylation degree increased to 10.76%, SEWP formed small and sparse aggregates; SEWP of a high succinylation degree (66.57%) formed small fibrous aggregates. This result illustrates that succinylation of low degree (2.70%–37.63%) can decrease the degree of cross-linking of SEWP and that SEWP with a higher succinylation degree (66.57%–73.84%) could form small fibrous aggregates. The electrostatic force is important in the formation of protein fibrous aggregates, and weak electrostatic repulsion contributes to the formation of fibrous aggregates [45]. The strong electrostatic repulsion and the surface hydrophobic interaction are balanced when there is a large amount of charge on the surface of a protein, thus forming fibrous aggregates. The formation of fibrous aggregates of a high succinylation degree (66.57%–73.84%) is consistent with the reduction of the zeta potential at a high succinylation degree in the section on zeta potential. This result indicates that succinylation actually prevented the thermal aggregation and improved the thermal stability of EWP. Succinylation destroyed the structure of EWP and changed the aggregated morphology of EWP.

## 3. Materials and Methods

### 3.1. Materials

Fresh eggs were purchased from Deqingyuan Co., Ltd. (Beijing, China). All of the chemical reagents were of analytical grade.

### 3.2. Preparation of Freeze-Dried EWP Powder

The preparation of freeze-dried EWP powder was according to [4]. Fresh egg white was diluted with double the volume of ultrapure water. The solution was gently stirred by a magnetic stirrer at 4 °C for 1 h. The egg white solution was dialyzed for 48 h while using a 1000 MW interception dialysis membrane at 4 °C and the water was changed every 12 h. The purpose of dialysis was to remove small molecules and salts. The samples were centrifuged at 8000× *g* for 30 min to remove large undissolved constituents (Centrifuge 5810, Eppendorf, Germany). The supernatant liquid was freeze-dried and then stored at −20 °C in a refrigerator.

### 3.3. Succinylation of EWP

The method of Wan et al. [18] was followed for preparing succinylated egg white protein (SEWP). The EWP solution was prepared by dispersing the EWP powder in ultrapure water (3.0% w/v) and adjusted to pH 8.0 while using 1.0 M NaOH solution. It was slowly stirred at room temperature for 1 h. Succinic anhydride was added in different additives at 0.25%, 0.5%, 1%, 5%, 10%, 15%, 20%, and 25%, relative to the EWP weight to the EWP solution, which was then adjusted to pH 8.0 with 0.1 M and 1 M NaOH solutions. After the stabilization of pH, it was stirred slowly at room temperature for 2 h. For dialysis of the prepared SEWP solution, we used a 1000 MW interception dialysis membrane and against ultrapure water at 4 °C for 48 h, with the water being changed every 12 h. The supernatant solution was lyophilized and stored at −20 °C. The control was prepared via the same above operation without the addition of succinic anhydride.

### 3.4. The Determination of Succinylation Degree

The determination of succinylation degree was carried out according to the method that was proposed by [46]. Glacial acetic acid was added to a lithium hydroxide solution to prepare lithium acetate buffer (4 M). Two grams of ninhydrin and 0.3 g of reductive reductant were dissolved in 75 mL of Dimethyl sulfoxide. A lithium acetate buffer (25 mL) was then added to the mixture; nitrogen was released for 2 min and then sealed with paraffin. The solution was stored at 4 °C. EWP powder (5 mg) was dissolved in 0.1 M NaOH solution (5 mL), 1 mL of samples and 1 mL of deionized water were transferred to a tube, and 2 mL of prepared ninhydrin solution was transferred to the tube. The solution was boiled for 15 min and then cooled to room temperature in an ice-water bath. Afterwards, the tubes were opened, 6 mL of 50% ethanol solution was added, and the mixture was mixed with a vortex mixer for 30 s. The absorbance of the mixture at 570 nm was determined with a UV spectrophotometer, with L-lysine hydrochloride being used as a standard. The degree of succinylation was calculated by the following formula:

Degree of succinylation% = (A − B)/A × 100 (1)

A = micromoles of free amino groups estimated per milligram of net protein (native)

B = micromoles of free amino groups estimated per milligram of net protein (succinylated)

### 3.5. Turbidity

EWP suspensions (5 mg/mL) with different succinylation degrees were prepared by dissolving EWP and SEWP powder in phosphate buffer (0.01 M, pH = 7.2). EWP and SEWP suspension (5 mg/mL) with various ionic strengths were prepared by adding various amounts of NaCl to the EWP and SEWP suspensions. The EWP/SEWP solutions (5 mg/mL) were heated at 90 °C for 30 min and then cooled quickly in an ice bath to room temperature. The absorbance of the mixture at 540 nm was determined to measure the turbidity of both dispersions.

### 3.6. Differential Thermal Analysis

The thermal stability of the samples was measured on a differential scanning calorimeter (Q200, Waters Corp., Milford, MA, USA). The EWP suspensions (100 mg/mL) with different succinylation degrees were prepared by dissolving EWP powder in phosphate buffer (0.01 M, pH = 7.2). The 10 μL samples were placed in an aluminum box, which was then hermetically sealed. The samples were heated from 20 to 110 °C at a rate of 10 °C min^−1^. A sealed empty aluminum box was used as a reference. The analysis of thermograms was performed while using TA Universal Analysis 2000 software.

### 3.7. Fourier Transform Infrared (FTIR) Spectroscopy

FTIR spectra were obtained by using a Thermo Fourier infrared spectrometer (Nicolet iS10). The samples were prepared by mixing EWP with dried KBr powder and then pressed into slices. The spectra were measured between 400 and 4000 cm^−1^ at a resolution of 4 cm^−1^.

### 3.8. Surface Hydrophobicity

EWP and SEWP were mixed with phosphate buffer saline (0.01 M, pH = 7.2) and then diluted to 0.0%–0.1% (*w*/*w*). ANS solution (25 μL, 0.008 M ANS, 0.01 M PBS) was added to 4 mL of the mixture, which was then mixed with a vortex mixer for 30 s and kept in a dark room for 15 min at room temperature. The fluorescence intensity was measured on a fluorospectrophotometer (F-7000, Hitachi Corp., Tokyo, Japan) at an excitation wavelength of 390 nm and an emission wavelength of 470 nm. Slit width was set to 5 nm. The fluorescence intensity of the buffer was subtracted to correct for the background fluorescence.

### 3.9. Intrinsic Fluorescence Spectroscopy

Intrinsic fluorescence spectroscopy was done with a Hitachi (H7000) fluorescence spectrometer. The protein solution (0.2 mg/mL) was prepared by dissolving EWP and SEWP in phosphate buffer solution (0.01 M, pH = 7.20). The excitation wavelength was 290 nm. The emission wavelength was scanned from 300 to 360 nm. Scanning speed was 240 nm min^−1^ and the excitation and emission slits widths were 2.5 nm for the determination. A series of fluorescence intensity measurements of gradient concentration were carried out in pretest to exclude the inner filter effect.

### 3.10. Zeta Potential

EWP and SEWP suspensions (5 mg/mL) for the determination of zeta potential were prepared by dissolving EWP and SEWP powder in phosphate buffer (0.01 M, pH = 7.2). The zeta potential at room temperature (26 °C) was measured by a Zetasizer (Nano ZS; Malvern Instruments Ltd. Malvern, WR14 1XZ, Worcestershire, UK) with a balance time of 2 min. Three measurements were performed for the average of each sample.

### 3.11. Transmission Electron Microscopy (TEM)

The TEM images were obtained with a transmission electron microscope (H-7500; Hitachi, Tokyo, Japan) at an acceleration voltage of 80 kV. SEWP and EWP suspensions (5 mg/mL) were prepared by dissolving EWP and SEWP powder in phosphate buffer (0.01 M, pH = 7.2), The EWP solution (5 mg/mL) was heated at 90 °C for 30 min and then cooled quickly in an ice bath to room temperature. The heated suspension of EWP and SEWP was diluted 50-fold with phosphate buffer (0.01 M, pH = 7.2). Thirty milliliters of diluted samples was transferred to a sealing membrane and the samples were negatively stained with 30 μL of 2% (*w*/*v*) silicon tungstate solution. Four-microliter samples were placed on a copper screen with a carbon-coated hydrophilic film. The copper screen was air-dried at room temperature for one day.

### 3.12. Statistical Analysis

Three replicates of all the measurement were conducted. The average and standard deviation were analyzed with Microsoft Excel 2007 (Microsoft Corp., Redmond, WA, USA). A significant-difference analysis was performed while using SPSS software (SPSS Inc., ver. 19, Armonk, IL, USA) and one-way analysis of variance was used to determine the statistical difference. Significant differences between means were identified while using Duncan’s multiple range tests (*p* < 0.05). All of the reported *p* values are two-sided and not adjusted for multiple testing.

## 4. Conclusions

Succinylation improved the thermal stability of EWP. After succinylation, the turbidity of the EWP solution (5 mg/mL) after heat treatment (90 °C, 30 min) significantly decreased. In contrast to the highly cross-linked aggregates of EWP, SEWP formed disperse aggregates after heat treatment. The introduction of negatively charged succinic anhydride groups increases the electrostatic repulsive force between EWP and colloidal stability. Succinylation with a low degree changed the conformational structure of EWP slightly, thus burying the hydrophobic groups that are primarily exposed on the protein surface in the interior regions and decreasing the surface hydrophobicity of EWP. The conformational structure of succinylated EWP with a high succinylation degree drastically changed and further formed fibrous aggregates with a high solubility. Succinylation effectively prevented thermal aggregation and improved the thermal stability of protein molecules. This study could expand the application of EWP in food manufacturing and provide theoretical support for the improvement of the thermal stability of EWP. The thermal aggregation and gel properties of SEWP could be further explored.

## Figures and Tables

**Figure 1 molecules-24-03783-f001:**
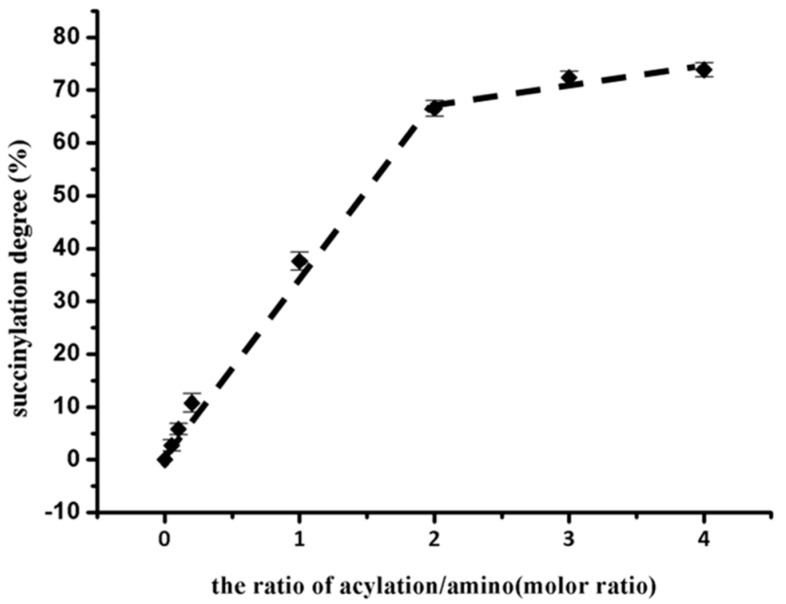
The relationship between the succinylation degree and various anhydrides: amino group molar ratios.

**Figure 2 molecules-24-03783-f002:**
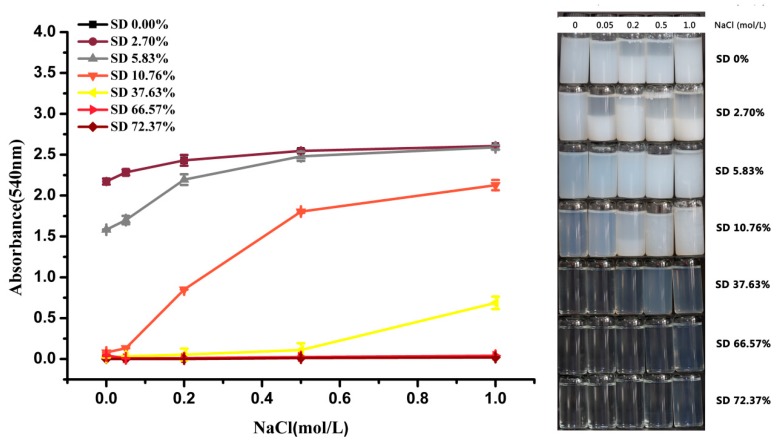
Changes in turbidity of egg white protein (EWP) and succinylated egg white protein (SEWP) solution (5 mg/mL) heated at 90 °C for 30 min at various NaCl concentrations (0.0–1.0 mol/L).

**Figure 3 molecules-24-03783-f003:**
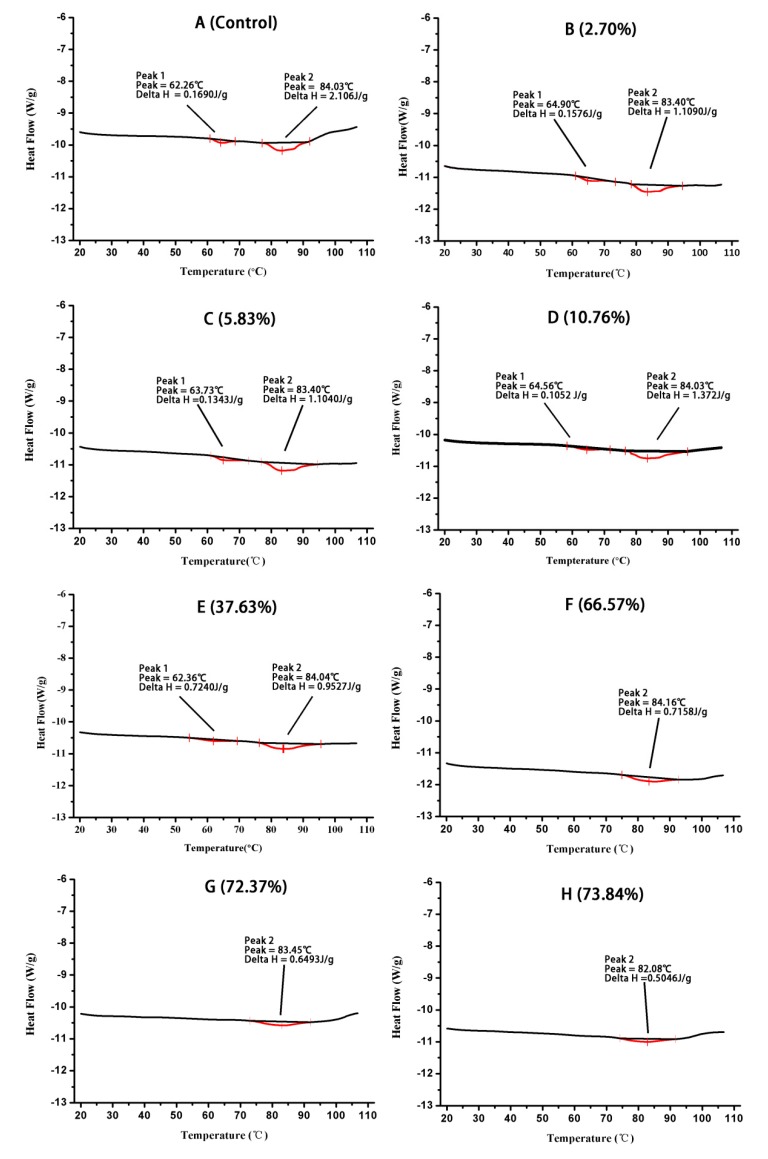
Differential Scanning Calorimetry (DSC) thermograms of EWP (**A**—0%) and SEWP with various succinylated degrees (**B**—2.70%, **C**—5.83%, **D**—10.76, **E**—37.63%, **F**–66.57%, **G**—72.37%, and **H**—73.84%).

**Figure 4 molecules-24-03783-f004:**
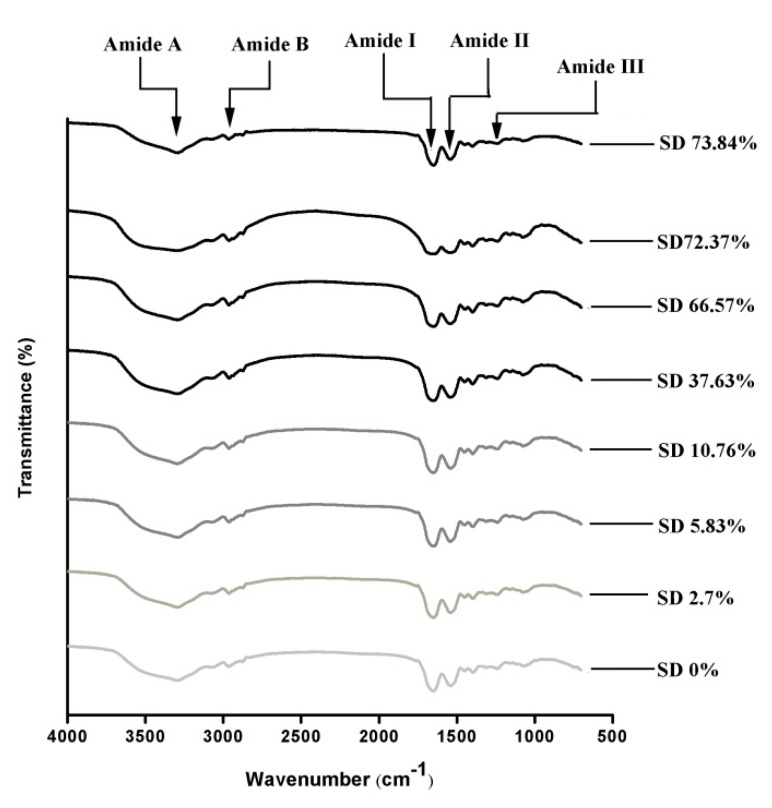
Fourier Transform Infrared (FTIR) spectra of samples of EWP (0%) and SEWP with various succinylation degrees (2.7%, 5.83%, 10.76%, 37.63%, 66.57%, 72.37%, and 73.84%).

**Figure 5 molecules-24-03783-f005:**
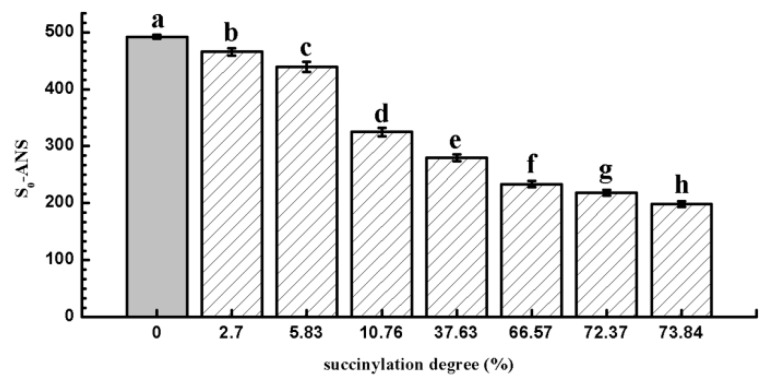
Changes in S_0_-ANS of EWP and SEWP (mean ± SD, *n* = 3). The different letters (a–h) above the columns indicate significant differences (*p* < 0.05) among the samples treated according to different succinylation degrees.

**Figure 6 molecules-24-03783-f006:**
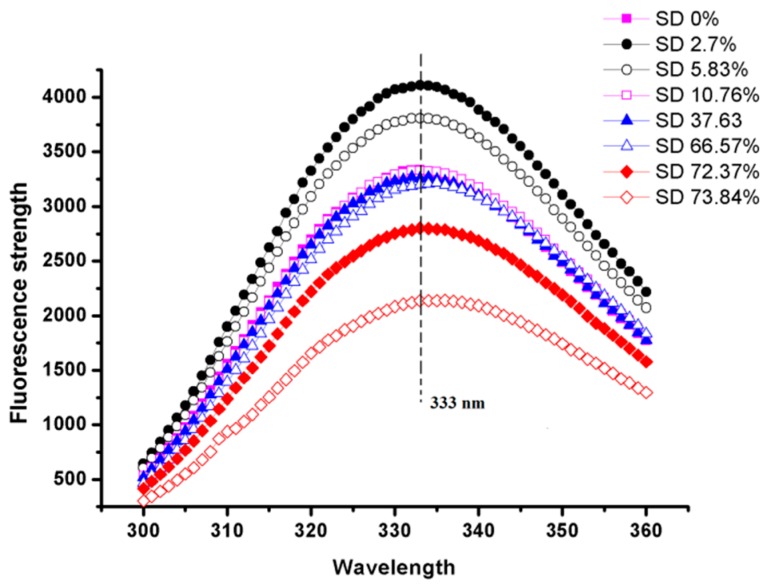
Intrinsic fluorescence spectra of EWP (0%) and SEWP with different succinylation degrees (2.7%, 5.83%, 10.76%, 37.63%, 66.57%, 72.37%, 73.84%).

**Figure 7 molecules-24-03783-f007:**
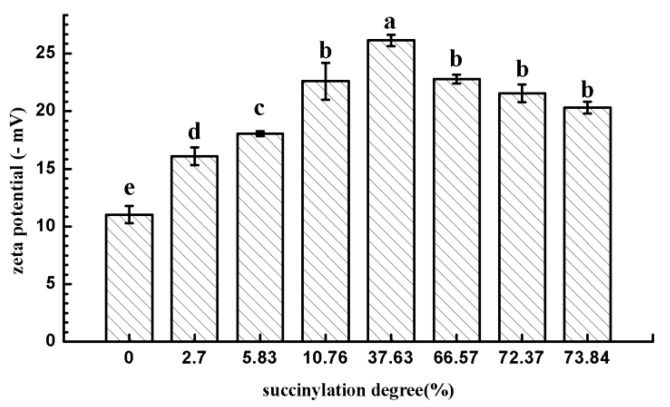
Changes in the zeta potential of EWP and SEWP (mean ± SD, *n* = 3). The different letters (a–e) above the columns indicate significant differences (*p* < 0.05) among the samples treated according to different succinylation degrees.

**Figure 8 molecules-24-03783-f008:**
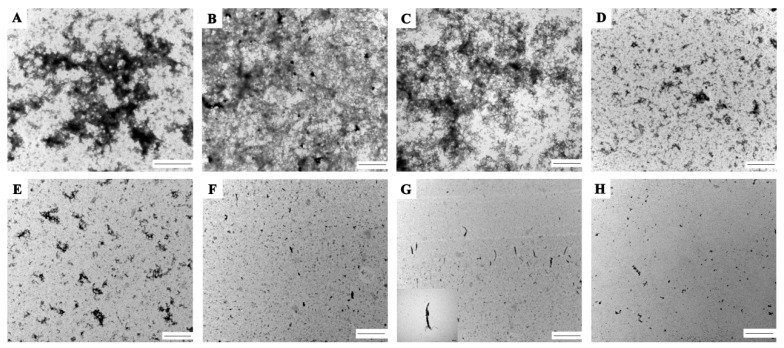
Representative Transmission Electron Microscopy (TEM) micrographs at 6000× of EWP and SEWP aggregates: A (control), B (2.70%), C (5.83%), D (10.76%), E (37.63%), F (66.57%), G (72.37%), and H (73.84%). All the samples were analyzed by TEM after heat treatment at 90 °C or 30 min. The scale bars represent 1 μm.

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
