# Peer review of "Succinylation Improves the Thermal Stability of Egg White Proteins"

_molecules, 2019, doi:10.3390/molecules24203783_

Round 1
Reviewer 1 Report
Comments to the Author
The objective of the article of “Succinylation improves the thermal stability of egg 2 white proteins” is interesting. However, there are some clear possibilities for improvement of the manuscript.
Major comments:
I would recommend that the authors get some help with respect to language editing. The current version is not too bad, but can still be improved, both with respect to language in general and with respect to the general presentation of the research. Author could compare the advantages of succinylation than other model organisms for thermal stability of egg white proteins. Because in title authors declare succinylation improves the thermal stability. Describe a little about succinylation before going to apply it. Discuss a little more statistical significance of proposed model? what does it means “These results are consistent with the changes in ΔH and Td of succinylated lysozyme, soy proteins, and yak casein micelles in a similar study “. Discuss a little more result sections. Figure 3 Y-axis level should be same. Please work on here. Description of Figures 4 and 6 also are not clear. In Figures 5 and 7, x-axis should use sequential length of succinylation degres ? Explain a little more. Some of the sentences are overestimated “The thermal aggregation and gel properties of SEWP must be explored “. Check all in the manuscript.Author Response
Dear reviewer,
We are truly grateful for yours and reviewers’ critical comments and thoughtful suggestions on our manuscript (Succinylation improves the thermal stability of egg white proteins). Based on these comments and thoughtful suggestions, we have made careful modifications on the original manuscript. All changes made to text are in red color. Because of your suggestions, the revised articles are better and readers can get more valuable information. We hope the new manuscript will meet your magazine’s standard. Below you will find our point-by-point responses to reviewers’ comments and questions.
Comment 1: I would recommend that the authors get some help with respect to language editing. The current version is not too bad, but can still be improved, both with respect to language in general and with respect to the general presentation of the research.
Response 1: We have improved the English writing by MDPI English editing service in the revised manuscript as suggested.
Comment 2: Author could compare the advantages of succinylation than other model organisms for thermal stability of egg white proteins. Because in title authors declare succinylation improves the thermal stability.
Response 2: At present, there were few studies for improving the thermal stability of EWP. In the introduction, we introduce two methods to improve the thermal stability of egg white in the other studies. “The addition of arginine at 106 mg/mL can effectively control the formation of insoluble aggregates of EWP solution (1 mg/ml)”, “high-pressure treatment can have negative effects on the rheological properties and color”, and high pressure treatment have problems with complex operation. So we add the following sentence in the introduction. ”Compared with the addition of arginine, succinylation process the advantages of high efficiency; and with the method of high pressure treatment, succinylation will be more practicable and effective.”
Comment 3: Describe a little about succinylation before going to apply it
Response 3: As suggested by reviewer we added the description in the section “Determination of Succinylation Degree” as follows. “Succinylation is a frequently used chemical modification method for protein and well known to improve functionality of proteins. Succinylation involves chemical derivatization of ε-amino group of lysine in protein with succinic anhydride”
Comment 4: Discuss a little more statistical significance of proposed model?
Response 4: As suggested by reviewers, we discussed more about the statistical significance: “Three replicates of all measurement were conducted.” “All reported p values are two-sided and were not adjusted for multiple testing.”
Comment 5: What does it means “These results are consistent with the changes in ΔH and Td of succinylated lysozyme, soy proteins, and yak casein micelles in a similar study “.
Response 5: We apologize for this ambiguous statement and we have adjusted to the text be clear. “These results are consistent with lysozyme, soy proteins, and yak casein micelles, with increasing succinylation, the Td value and ΔH of these proteins decreased.”
Comment 6: Discuss a little more result sections.
Response 6: as suggested by reviewer, we have added more discussion.
Comment 7: Figure 3 Y-axis level should be same. Please work on here.
Response 7: We apologize for this error and unify the Y-axis as suggested.
Comment 8: Description of Figures 4 and 6 also are not clear.
Response 8: In the section of “Determination of Succinylation Degree”, Succinic anhydride were added in different additives at 0.25%, 0.5%, 1%, 5%, 10%, 15%, 20%, and 25% relative to the EWP weight to the EWP solution, and obtained egg white proteins with various succinylated degrees (2.70%, 5.83%, 10.76, 37.63%, 66.57%, 72.37%, 73.84%). So we used the results of the section of “Determination of Succinylation Degree” as the x-axis.
Comment 9: Some of the sentences are overestimated “The thermal aggregation and gel properties of SEWP must be explored “. Check all in the manuscript.
Response 9: We apologize for these overestimated statements, so we check the manuscript and revised: “The thermal aggregation and gel properties of SEWP could be further explored.” “We hypothesized that succinylation could increase the thermal stability of EWP.” “This phenomenon probably resulted from changes in the structures of soy proteins and casein due to the exposure of lysine groups buried in the interior regions of the protein upon the induction of succinylated groups with further succinylation [18], but with the increase in the amount of added succinic anhydride, there was almost no further exposure of lysine groups on the surface of EWP molecules.” “This further illustrated that succinylation of low degree changed the structure of EWP slightly and that succinylation of high degree destroyed drastically the structure of EWP.”
Reviewer 2 Report
This work analyzes the effect of Succinylation on thermal stability of egg white protein”, these subject it is of interest to food industry the discussion of results obtained is confused and several discrepancies are observed.
Page 2 line 60-63 , the authors argue “Succinylation promotes the dissociation of
phosphoprotein of yak casein micelles and converts the ordered secondary structure of protein to disordered structure, thus lowering the surface hydrophobicity and the heat stability of yak casein micelles” however, Yang et al, reference [15], concluded that “With increasing succinylation, turns decreased, ß-sheet and α-helix increased, and irregular structure were non-significantly affected.” . How do you explain this discrepancy?
Page 4 line 133: the authors argue that “after succinylation of high degree, ovotransferrin did not denature and denatured ovalbumin decreased” because succinylation (73.8%) led to the disappearance of peak 1, and the area of peak 2 decreased remarkably. But this behavior can be explained considering that ovotransferrin (totally) and ovoalbumin (partially) are denatured beforehand.
Page 7
line 193-195: Please indicate what it means “the fluorescence quenching factor”. What is the quencher that does not perform its quencher function when succinylation changed the structure of EWP? In reference [43] there is no mention about this effect of increased flourescence.
lines 196-205: The authors argue with increase of SD (66.57-74.83%) “λ max of SEWP increased from 333 nm to 336 nm, red shifts occurred, and the
fluorescence intensity decreased. This phenomenon illustrates that the polarity of the
microenvironment of the tryptophan residue increased and that succinylation destroyed the surface hydrophobic interaction within protein molecules” this behavior shows that the fluorophores are exposed at solvent because “the protein unfolded, and its structure collapsed”, however in conclusion section the authors argue (lines 340-341) “Succinylation changed the conformational structure of EWP, thus burying the hydrophobic groups that are primarily exposed on the protein surface in the interior regions”
Page 8: It is hardly to see the maxima of fluorescence in figure 6.
Section 3.8: The inner filter effect was considered and corrected?
Minor revision
Page 1 line 39. The sentence must begin “Hong et al [4]”
Please describe with more detail the section 3.3
Page 5: Please, add the succinylation degree in figure caption
In section 3.4 there is no any information about NaCl solution used for obtain the results showed in figure 2.
Page 10 lines 276- 277, Please reformulate the sentence “Different additives of succinic anhydride (0.25%, 0.5%, 1%, 5%, 10%, 15%, 20%, and 25% relative to the EWP weight) were added to the EWP solution”
Author Response
Dear editors and reviewers,
We are truly grateful for yours and reviewers’ critical comments and thoughtful suggestions on our manuscript (Succinylation improves the thermal stability of egg white proteins). Based on these comments and thoughtful suggestions, we have made careful modifications on the original manuscript. All changes made to text are in red color. Because of your suggestions, the revised articles are better and readers can get more valuable information. We hope the new manuscript will meet your magazine’s standard. Below you will find our point-by-point responses to reviewers’ comments and questions.
Comment 1: Page 2 line 60-63, the authors argue “Succinylation promotes the dissociation of phosphoprotein of yak casein micelles and converts the ordered secondary structure of protein to disordered structure, thus lowering the surface hydrophobicity and the heat stability of yak casein micelles” however, Yang et al, reference [15], concluded that “With increasing succinylation, turns decreased, ß-sheet and α-helix increased, and irregular structure were non-significantly affected.” . How do you explain this discrepancy?
Response 1: We apologize for the inaccuracy of the summary of reference, as your suggestion we checked the content of this reference. We make changes as follows: “Succinylation promotes the dissociation of phosphoprotein of yak casein micelles and changed the secondary structure of protein, thus lowering the surface hydrophobicity and the heat stability of yak casein micelles [15, 16].”
Comment 2: Page 4 line 133: the authors argue that “after succinylation of high degree, ovotransferrin did not denature and denatured ovalbumin decreased” because succinylation (73.8%) led to the disappearance of peak 1, and the area of peak 2 decreased remarkably. But this behavior can be explained considering that ovotransferrin (totally) and ovalbumin (partially) is denatured beforehand.
Response 2: We appreciate the reviewer’s suggestions for the discussion of the result of DSC. In the process of succinylation, the pH of reaction system is maintained at about 8.0, we think the possibility of denaturation is low.
Comment 3: Line 193-195: Please indicate what it means “the fluorescence quenching factor”. What is the quencher that does not perform its quencher function when succinylation changed the structure of EWP? In reference [43] there is no mention about this effect of increased fluorescence intensity.
Response 3: Fluorescence quenching refers to any process which decreases the fluorescence emission intensity of a given substance. Fluorescence quenching factor refers to substance resulted in the fluorescence.
As suggested by the reviewer we checked the discussion and reference and revised as follows: This phenomenon could interpreted by a high content of tryptophan exposed in polar aqueous solution, leading to spontaneous fluorescence quenching with the introduction of succinic anhydride groups; the conformational structure of proteins changed and the amount of lysine exposed to solution decreased, which weakened the fluorescence quenching, thus, the fluorescence intensity increased [44].
Comment 4: lines 196-205: The authors argue with increase of SD (66.57-74.83%) “λmax of SEWP increased from 333 nm to 336 nm, red shifts occurred, and the fluorescence intensity decreased. This phenomenon illustrates that the polarity of the microenvironment of the tryptophan residue increased and that succinylation destroyed the surface hydrophobic interaction within protein molecules” this behavior shows that the fluorophores are exposed at solvent because “the protein unfolded, and its structure collapsed”, however in conclusion section the authors argue (lines 340-341) “Succinylation changed the conformational structure of EWP, thus burying the hydrophobic groups that are primarily exposed on the protein surface in the interior regions”
Response 4: As suggested by reviewers we revised the discussion and conclusion as follows: “During the preliminary stage of succinylation, amino acids on the surface of EWP reacted with succinic anhydride and the introduction of a small amount of negatively succinic anhydride group, resulting in a slight change in the conformational structure of EWP. In addition, tryptophan residues and surface hydrophobic groups on the protein surface became buried in the interior of the protein, making it difficult for the succinylated protein to aggregate. When the succinylation degree continued to increase, the introduction of a large amount of succinic anhydride group destroyed the interior structure of EWP and the protein structure changed dramatically.” “Succinylation with a low degree changed the conformational structure of EWP slightly, thus burying the hydrophobic groups that are primarily exposed on the protein surface in the interior regions and decreasing the surface hydrophobicity of EWP.”
Comment 5: Page 8: It is hardly to see the maxima of fluorescence in figure 6.
Response 5: As suggested by reviewer we added the 333 nm mark line in figure 6.
Comment 6: Section 3.8: The inner filter effect was considered and corrected?
Response 6: We have considered the inner filter effect. A series of fluorescence intensity measurements of gradient concentration were carried out and excluded the inner filter effect.
Comment 7: Page 1 line 39. The sentence must begin “Hong et al [4]”
Response 7: We apologize for this error and as suggested by reviewer corrected this error: “Hong et al. [4] reported that the addition of arginine at 106 mg/mL can effectively control the formation of insoluble aggregates of EWP solution (1 mg/ml).”
Comment 8: Please describe with more detail the section 3.3
Response 8: As suggested by reviewer we added the detail of the method in section 3.3. “The determination of succinylation degree was carried out according to the method proposed by [46]. Glacial acetic acid was added to a lithium hydroxide solution to prepare lithium acetate buffer (4 M). Two grams of ninhydrin and 0.3 g of reductive reductant were dissolved in 75 mL of Dimethyl sulfoxide. A lithium acetate buffer (25 mL) was then added to the mixture; nitrogen was released for 2 min and sealed with paraffin. The solution was stored at 4 °C. EWP powder (5 mg) was dissolved in 0.1 N NaOH solution (5 mL), 1 mL of samples and 1 mL of deionized water were transferred to a tube, and 2 mL of prepared ninhydrin solution was transferred to the tube. The solution was boiled for 15 min and then cooled to room temperature in an ice-water bath. Then, the tubes were opened, 6 mL of 50% ethanol solution was added, and the mixture was mixed with a vortex mixer for 30 s. The absorbance of the mixture at 570 nm was determined with a UV spectrophotometer, with L-lysine hydrochloride used as a standard. The degree of succinylation was calculated by the following formula:
Degree of succinylation % =(A—B)/A×100
A = micromoles of free amino groups estimated per milligram of net protein (native).
B = micromoles of free amino groups estimated per milligram of net protein (succinylated)”
Comment 9: Page 5: Please, add the succinylation degree in figure caption
Response 9: As suggested by reviewer we added the succinylation degree in figure caption. “Fig 3. Differential Scanning Calorimetry (DSC) thermograms of EWP (A—0%) and SEWP with various succinylated degrees (B—2.70%, C—5.83%, D—10.76, E—37.63%, F–66.57%, G—72.37%, H—73.84%)”
Comment 10: In section 3.4 there is no any information about NaCl solution used for obtaining the results showed in figure 2.
Response 10: As suggested by reviewer we added in information about NaCl solution in the method. “EWP and SEWP suspension (5 mg/mL) with various ionic strengths were prepared by adding various amounts of NaCl to the EWP and SEWP suspensions.”
Comment 11: Page 10 lines 276-277, Please reformulate the sentence “Different additives of succinic anhydride (0.25%, 0.5%, 1%, 5%, 10%, 15%, 20%, and 25% relative to the EWP weight) were added to the EWP solution”
Response 11: As suggested by review we reformulate the sentence. “Succinic anhydride was added in different additives at 0.25%, 0.5%, 1%, 5%, 10%, 15%, 20%, and 25%, relative to the EWP weight to the EWP solution, which was then adjusted to pH 8.0 with 0.1 M and 1 M NaOH solutions.”
Reviewer 3 Report
`In this work by He, Lv and Tong, the proposition that succinylation improves the thermal stability of EWP is made. However I have got an impression that either the authors failed to correctly present their data or they do not understand the limitations of certain techniques and miss some general principles of protein organization.
First of all, the authors should really describe their experimental procedure of succinylation and the way they estimated its success. The single line in the materials and methods is clearly not enough.
What is the reason that the molar ratio of 2.0 is the critical point of succinylation? Explanation is necessary.
Lines 85-86 the possible reason in different succinylation degree is proposed to be linked to the diversity in protein structure - but what do yo mean? All these proteins are globular - so structurally, in general, they are similar. If you mean different content of lysines (and better to say surface exposed lysines) it is a very different reason.
In line 91 it is stated that it was impossible to reach the succinylation to more than 75%, but the quick control experiment could be to denature protein first and then do succinylation, however this control is missing.
In the paragraph 2.2 about turbidity the authors confuse in my opinion different phenomena. From their description it seems that they are observing the salting out effect caused by the high concentration of added sodium chloride. many proteins are not stable in the high salt - but it has nothing to do with the thermal stability. I wonder how NaCl can decrease the amount of the hydrosulfonyl - it can perhaps shield its charge. So the experiment described in Fig 2 evaluates the ability of succinylation to prevent the precipitation of the protein in the high salt condition (in principle the authors should have tried even higher salt concentration) and not the thermal stability. Furthermore the error bars are missing in the figure.
For DSC experiments it is very hard to interpret the presented graphs, please produce the standard Cp graphs to easily visualise the changes in Tm - that's crucial to understand whether indeed succinylation provides thermal stability.
In line 149 the authors claim that conversion of alpha helix into beta sheet occurs - however it is hard to believe. The ideal control would be getting a structure of a succinylated protein (any of EWP) to demostrate it, but I assume the technique is not available to the authors. In this case at lease CD spectroscopy control should be done to show the decrease in the alpha helical content.
Line 160 - what kind of contacts do you mean bettwen -CH2 and side chains?
Line 162 - how can succinylation strengthen the hydrogen bonds? succinylation just leads to the swap of a positive charge on lysine side chain to the negative - so the authors should explore electrostatic changes.
In Lines 166 - 169 the authors make a suggestion that succinylation forces the exposed hydrophobic side chains to bury inside however this is the well known fact that globular proteins always bury their hydrophobic side chains into the so-called hydrophobic core and polar and charged side chains are located at the surface. One more time - succinylation leads to the change of the charge so the change in hydrophobicity is hard to envisage. Furthermore the probe ANS has a negatively charged sulfonic acid moiety and it is known that Lys side chains are potential binders of this dye. Hence what the authors observe in the figure 5 is simply the decrease in the amount of available binding sites for NAS
In lines 198-203 the authors again discuss rather impossible scenario of transfer of hydrophobic residues. In fact I think the authors observe the stabilization of proteins due to the acquired extra negative charge.
Line 251 - the authors should elaborate how weak electrostatic repulsions contribute to the formation of fibrous aggregates.
Author Response
Dear reviewer,
We are truly grateful for yours and reviewers’ critical comments and thoughtful suggestions on our manuscript (Succinylation improves the thermal stability of egg white proteins). Based on these comments and thoughtful suggestions, we have made careful modifications on the original manuscript. All changes made to text are in red color. Because of your suggestions, the revised articles are better and readers can get more valuable information. We hope the new manuscript will meet your magazine’s standard. Below you will find our point-by-point responses to reviewers’ comments and questions.
Comment 1: First of all, the authors should really describe their experimental procedure of succinylation and the way they estimated its success. The single line in the materials and methods is clearly not enough.
Response 1: As suggested by reviewer, we have added the procedure of succinylation to the section of methods. “Glacial acetic acid was added to a lithium hydroxide solution to prepare lithium acetate buffer (4 M). Two grams of ninhydrin and 0.3 g of reductive reductant were dissolved in 75 mL of Dimethyl sulfoxide. A lithium acetate buffer (25 mL) was then added to the mixture; nitrogen was released for 2 min and sealed with paraffin. The solution was stored at 4 °C. EWP powder (5 mg) was dissolved in 0.1 N NaOH solution (5 mL), 1 mL of samples and 1 mL of deionized water were transferred to a tube, and 2 mL of prepared ninhydrin solution was transferred to the tube. The solution was boiled for 15 min and then cooled to room temperature in an ice-water bath. Then, the tubes were opened, 6 mL of 50% ethanol solution was added, and the mixture was mixed with a vortex mixer for 30 s. The absorbance of the mixture at 570 nm was determined with a UV spectrophotometer, with L-lysine hydrochloride used as a standard. The degree of succinylation was calculated by the following formula:
Degree of succinylation % =(A—B)/A×100
A = micromoles of free amino groups estimated per milligram of net protein (native).
B = micromoles of free amino groups estimated per milligram of net protein (succinylated)
The way we estimated the improvement including the determination of turbidity and DSC, and the TEM was used to confirm the result of this experiment. In this study, we refer to some references which mainly used the turbidity and DSC to evaluate the improvement in thermal stability of proteins. (References: Arginine prevents thermal aggregation of hen egg white proteins. Effects of succinylation on the structure and thermostability of lysozyme.)
In this study, we used the freeze-drying egg white protein as the materials for succinylation. The purpose of this approach is to let the undenatured egg white as the substance and keep materials consistent. In follow-up studies, we will use various materials as substance and compare the differences.
Comment 2: What is the reason that the molar ratio of 2.0 is the critical point of succinylation? Explanation is necessary.
Response 2: As suggested by reviewer we added the reason in the discussion. “Succinylation proceeds from the surface of proteins to the inside of proteins; a critical succinylation degree indicates that the protein structure changed dramatically [19]. This result indicated that when the molar ratio raised to 2.0, the succinylation degree increased to 66.57%, indicating that the content of lysine in the surface of EWP was at 66.57% approximately, and with the further increase of molar ratio, the lysine exposed from the inside of proteins further reacted with succinic anhydride.”
Comment 3: Lines 85-86 the possible reason in different succinylation degree is proposed to be linked to the diversity in protein structure - but what do you mean? All these proteins are globular - so structurally, in general, they are similar. If you mean different content of lysine (and better to say surface exposed lysine) it is a very different reason.
Response 3: We apologize for this ambiguous statement. We think carefully reviewers’ comments and revised the discussion as follows. “We speculate that the difference in the succinylation degree was due to the diversity of type of proteins and the amount of lysine exposed to the surface of protein. Although the major component of soy protein, casein, and EWP is globular proteins [20-22], original casein micelles in milk were destroyed and separated with a mineral substance in the production of casein, which led to casein molecular unfolding and to the exposure of a large amount of lysine to react with succinic anhydride [23, 24]. Thus, the various content of lysine exposed to the surface of proteins probably led to the difference in the succinylation degree of proteins.”
Comment 4: In line 91 it is stated that it was impossible to reach the succinylation to more than 75%, but the quick control experiment could be to denature protein first and then do succinylation, however this control is missing.
Response 4: This is a valid question. In fact, we want to improve the thermal stability of egg white protein by succinylation and explore the mechanism of improving thermal stability. In the follow-up study, we will supplementary the quick control experiment.
Comment 5: In the paragraph 2.2 about turbidity the authors confuse in my opinion different phenomena. From their description it seems that they are observing the salting out effect caused by the high concentration of added sodium chloride. Many proteins are not stable in the high salt - but it has nothing to do with the thermal stability. I wonder how NaCl can decrease the amount of the hydrosulfonyl - it can perhaps shield its charge. So the experiment described in Fig 2 evaluates the ability of succinylation to prevent the precipitation of the protein in the high salt condition (in principle the authors should have tried even higher salt concentration) and not the thermal stability. Furthermore the error bars are missing in the figure.
Response 5: We considered carefully about the opinion of reviewer. Before heat treatment, the added NaCl not lead the egg white protein salting out according our observation. The changes of turbidity producing after the heat treatment, so we think this precipitation have some relation with the heat treatment and thermal stability. Succinylation prevent ion-induced aggregation of egg white protein. As suggested by reviewer, we changed “This result suggests that succinylation can prevent ion-induced aggregation and improve the thermal stability of EWP.” to “This result suggests that succinylation can prevent ion-induced aggregation of EWP.”
We apologize for the incorrect statement of the “hydrosulfonyl”. We rereading the reference and revised the discussion as follow. “The addition of NaCl can decrease the amount of hydrosulfonyl exposed on the surface of protein in the process of heat treatment, increase the surface hydrophobicity, and neutralize the charge on the protein surface.”
As suggested by reviewer we have added error bars in the figure.
Comment 6: For DSC experiments it is very hard to interpret the presented graphs, please produce the standard Cp graphs to easily visualise the changes in Tm - that's crucial to understand whether indeed succinylation provides thermal stability.
Response 6: We apologize for this nonstandard. We determined the DSC experiment in the other institution and we get the analysis results by the analysis software “Universal V4.7A TA Instruments”. However, the pictures we got were not suitable to apply to manuscript because of thin line and picture format, so we used the raw date plotting by the origin 8.0 and marked the Td and Delta H.
Comment 7: In line 149 the authors claim that conversion of alpha helix into beta sheet occurs - however it is hard to believe. The ideal control would be getting a structure of a succinylated protein (any of EWP) to demonstrate it, but I assume the technique is not available to the authors. In this case at lease CD spectroscopy control should be done to show the decrease in the alpha helical content.
Response 7: We agree with reviewers’ opinion about the determination of the secondary structure by the CD spectroscopy. In addition, we determined the CD spectroscopy and repeated it in the primary of this study. The results of this determination have different from the previous studies. So we will repeat and confirm it in the next section of experiment and contrastive analysis the differences.
Comment 8: Line 160 - what kind of contacts do you mean between -CH2 and side chains?
Response 8: We apologize for this ambiguous statement. This contacts refer to the asymmetric stretching vibration of the –CH2 group as well as NH3+ of peptide side chains. So we revised the discussion as follows. “This shift illustrates the interaction between NH3+ of the peptide chain and the asymmetric stretching vibration of the –CH2 group as well as NH3+ of peptide side chains”
Comment 9: Line 162 - how can succinylation strengthen the hydrogen bonds? Succinylation just leads to the swap of a positive charge on lysine side chain to the negative - so the authors should explore electrostatic changes.
Response 9: We apologize for this ambiguous statement. In the section of FTIR spectra, changes in structure of succinylated egg white protein were explored. But as a whole succinylation induce the negative charge to the amino acid, thus changed the structure of egg white proteins, such as hydrogen bonds. And we make a discussion about the electrostatic changes in the section of Zeta Potential and conclusion.
Comment 10: In Lines 166 - 169 the authors make a suggestion that succinylation forces the exposed hydrophobic side chains to bury inside however this is the well known fact that globular proteins always bury their hydrophobic side chains into the so-called hydrophobic core and polar and charged side chains are located at the surface. One more time - succinylation leads to the change of the charge so the change in hydrophobicity is hard to envisage. Furthermore the probe ANS has a negatively charged sulfonic acid moiety and it is known that Lys side chains are potential binders of this dye. Hence what the authors observe in the figure 5 is simply the decrease in the amount of available binding sites for ANS.
Response 10: We consider reviewers’ comments seriously.
Although the globular proteins bury their hydrophobic side chains into the so-called hydrophobic core, there are some hydrophobic amino acids exposed in the surface of protein, such as phenylalanine and valine. And there are some studies about succinylation of proteins, and the surface hydrophobicity was determined to illustrate the mechanism of succinylation. The references are as follows (Effect of succinylation on the physicochemical properties of soy protein hydrolysate. Effects of succinylation on the structure and thermal aggregation of soy protein isolate. Acetylation and succinylation of faba bean legumin: Modification of hydrophobicity and conformation.)
When the pH is about 7.4 the measured surface hydrophobicity is reliable, and ANS did not bind to the positively charged residues on the protein surface. In the study, the determination system of surface hydrophobicity is buffer solution with pH=7.2.
As suggested by reviewer we revised the manuscript as follows. “The amount of available binding sites for ANS decreased and less of the ANS probe could access and bind to hydrophobic groups.”
Comment 11: In lines 198-203 the authors again discuss rather impossible scenario of transfer of hydrophobic residues. In fact I think the authors observe the stabilization of proteins due to the acquired extra negative charge.
Response 11: As suggested by reviewer we revised the discussion as follows. “During the preliminary stage of succinylation, amino acids on the surface of EWP reacted with succinic anhydride, the introduction of a small amount of negatively succinic anhydride group resulting in a slight change in the conformational structure of EWP. And tryptophan residues and surface hydrophobic groups on the protein surface became buried in the interior of the protein, making it hard for the succinylated protein to aggregate. When the succinylation degree continued to increase, introduction of a large amount of succinic anhydride group destroyed the interior structure of EWP, the protein structure changed dramatically.”
Comment 12: Line 251 - the authors should elaborate how weak electrostatic repulsions contribute to the formation of fibrous aggregates.
Response 12: As suggested by reviewer we revised the manuscript as follows. “When there has a large amount of charge in the surface of protein, strong electrostatic repulsion and surface hydrophobic interaction will be balanced, thus forming fibrous aggregates.”
Round 2
Reviewer 1 Report
Until now, the RSCP is the most updated computational predictor and available for the identification of RSCs [6]. Therefore, to compare the proposed predictor with the RSCP, the same training and independent dataset were collected from the RSCP predictor. All the curated sequences were repossessed from Uniprot (https://www.uniprot.org/). The collected sequences were mainly from mammals, algae, plants, yeast, microbes, and diverse parasites. In training datasets, we reserved 456 protein sequences with 758 RSCs (positive samples). The other cysteines that were not identified as RSCs were regarded as redox-insensitive cysteine samples (negative samples). The independent dataset consisting of 99 proteins with 160 positive and 376 negative samples were retrieved from the RSCP predictor. In the training model, we considered a balanced dataset (i.e. positive: negative = 1:1). In the independent dataset, all the negative and positive samples were recalled to simulate the real situation. A sequence window with 41 amino acid residues was considered.
Author Response
Dear editor and reviewers,
We are truly grateful for yours and reviewers’ critical comments and thoughtful suggestions on our manuscript (Succinylation improves the thermal stability of egg white proteins). Because of your suggestions, the revised articles are better and readers can get more valuable information. We hope the new manuscript will meet your magazine’s standard.
Reviewer 2 Report
The work was improved with the corrections that were made. However, there are still some issues that were not considered.
Page 4 line 149: the authors argue that “after succinylation of high degree, ovotransferrin did not denature and denatured ovalbumin decreased” because succinylation (73.8%) led to the disappearance of peak 1, and the area of peak 2 decreased remarkably. But this behavior can be explained considering that ovotransferrin (totally) and ovoalbumin (partially) are denatured beforehand.
The behavior observed in figure 6 could be caused by an internal filter effect that in section 3.8 does not indicate whether or not it was considered.
Author Response
Dear editor and reviewers,
We are truly grateful for yours and reviewers’ critical comments and thoughtful suggestions on our manuscript (Succinylation improves the thermal stability of egg white proteins). Based on these comments and thoughtful suggestions, we have made careful modifications on the original manuscript. We apologize for the ambiguous explanation for reviewer. All changes made to text are highlighted. Because of your suggestions, the revised articles are better and readers can get more valuable information. We hope the new manuscript will meet your magazine’s standard. Below you will find our point-by-point responses to reviewers’ comments and questions.
Comment 1: Page 4 line 149: the authors argue that “after succinylation of high degree, ovotransferrin did not denature and denatured ovalbumin decreased” because succinylation (73.8%) led to the disappearance of peak 1, and the area of peak 2 decreased remarkably. But this behavior can be explained considering that ovotransferrin (totally) and ovoalbumin (partially) are denatured beforehand.
Response 1: We apologize for ambiguous explanation in the last revision, and we considered the comment seriously. As suggested by reviewer we revised the manuscript as follows. “This indicates that after succinylation of a high degree, ovotransferrin have denatured totally and ovalbumin have denatured partially beforehand, which probably related to the changes in conformational structure of EWP after succinylation of high degree.”
Comment 2: The behavior observed in figure 6 could be caused by an internal filter effect that in section 3.8 does not indicate whether or not it was considered.
Response 2: We apologize for not revising the mistake in the last revision and just making explanation in cover letter. So as suggested by reviewer we revised the manuscript as follows. “A series of fluorescence intensity measurements of gradient concentration were carried out in pretest to exclude the inner filter effect.”
Reviewer 3 Report
The authors modified their manuscript and answered most of the concerns.
I leave to an editor to decide about whether the revised version is suitable now for publication in Molecules.
Do one more round of proff-reading, I noticed some typos, like 0.1 N NaOH, should be 0.1 M NaOH, and so on
Author Response
Dear editor and reviewers,
We are truly grateful for yours and reviewers’ critical comments and thoughtful suggestions on our manuscript (Succinylation improves the thermal stability of egg white proteins). Based on these comments and thoughtful suggestions, we have made careful modifications on the original manuscript. We apologize for this mistake. All changes made to text are highlighted. Because of your suggestions, the revised articles are better and readers can get more valuable information. We hope the new manuscript will meet your magazine’s standard. Below you will find our point-by-point responses to reviewers’ comments and questions.
Comment 1: Do one more round of proff-reading, I noticed some typos, like 0.1 N NaOH, should be 0.1 M NaOH, and so on
Response 1: As suggested by reviewer we made proff-reading and revised the manuscript as follows. “The solution was stored at 4 °C. EWP powder (5 mg) was dissolved in 0.1 M NaOH solution (5 mL), 1 mL of samples and 1 mL of deionized water were transferred to a tube, and 2 mL of prepared ninhydrin solution was transferred to the tube.” “The purpose of dialysis was to remove small molecules and salts. The samples were centrifuged at 8000´ g for 30 min to remove large undissolved constituents (Centrifuge 5810, Eppendorf, Germany).” “Compared with the EWP, the turbidity of SEWP decreased significantly (p < 0.05).” “The 10 μL samples were placed in an aluminum box, which was then hermetically sealed. Samples were heated from 20 to 110 °C at a rate of 10 °C min-1.”